# Day-to-day variability of knee pain and the relationship with physical activity in people with knee osteoarthritis: an observational, feasibility study using consumer smartwatches

Arani Vivekanantham [1,2,3] David Selby,[1] Mark Lunt,[1] Jamie C Sergeant,[1] Matthew J Parkes [1] Terence W O'Neill,[1,4,5] Will Dixon[1,4,5]

For numbered affiliations see end of article.

**Correspondence to**
Dr Arani Vivekanantham;
arani.vivekanantham@ndorms.ox.ac.uk

## ABSTRACT

**Objective** To assess the feasibility of using smartwatches in people with knee osteoarthritis (OA) to determine the day-to-day variability of pain and the relationship between daily pain and step count.

**Design** Observational, feasibility study.

**Setting** In July 2017, the study was advertised in newspapers, magazines and, on social media. Participants had to be living/willing to travel to Manchester. Recruitment was in September 2017 and data collection was completed in January 2018.

**Participants** 26 participants aged≥50 years with self-diagnosed symptomatic knee OA were recruited.

**Outcome measures** Participants were provided with a consumer cellular smartwatch with a bespoke app that triggered a series of daily questions including two times per day questions about level of knee pain and one time per month question from the pain subscale of the Knee Injury and Osteoarthritis Outcome Score (KOOS) questionnaire. The smartwatch also recorded daily step counts.

**Results** Of the 25 participants, 13 were men and their mean age was 65 years (standard deviation (SD) 8 years). The smartwatch app was successful in simultaneously assessing and recording data on knee pain and step count in real time. Knee pain was categorised into sustained high/low or fluctuating levels, but there was considerable day-to-day variation within these categories. Levels of knee pain in general correlated with pain assessed by KOOS. Those with sustained high/low levels of pain had a similar daily step count average (mean 3754 (SD 2524)/4307 (SD 2992)), but those with fluctuating pain had much lower step count levels (mean 2064 (SD 1716)).

**Conclusions** Smartwatches can be used to assess pain and physical activity in knee OA. Larger studies may help inform a better understanding of causal links between physical activity patterns and pain. In time, this could inform development of personalised physical activity recommendations for people with knee OA.

## STRENGTHS AND LIMITATIONS OF THIS STUDY

⇒ Our app was successful in assessing and recording data on patient-reported symptoms and continuous sensor data simultaneously in real time.

⇒ As this was a feasibility study, participant numbers were relatively small.

⇒ Subjects volunteered for participation, and it is possible that selection factors and in particular an interest in smartwatches may have influenced engagement and reporting, limiting the generalisation of the findings.

⇒ Participants self-reported their knee osteoarthritis (OA); participants may have had varying severities of their knee OA (early disease to end-stage).

⇒ The step count used was an estimate of the true step count levels, which was broken up by charging cycles meaning that some activity may have been missed leading to measurement error; also, the total step count is a very rough measure of physical activity and misses patterns within the day.

## INTRODUCTION

Knee osteoarthritis (OA) affects one in eight people over the age of 60 years and pain is a cardinal symptom. Longitudinal studies suggest that the symptoms of knee OA typically follow relatively stable long-term trajectories;[1 2] however, this has been based on infrequently collected data. This is because clinical practice and most research studies capture pain reports at discrete time points, often months apart.[3] We thus have a limited picture of daily pain trajectories; such data are important because variations in pain, and its inherent uncertainty and unpredictability, can impact significantly on patients' quality of life. It has more recently been proven that there is substantial day-to-day and within-day variability of pain in people with knee OA.[4 5] For example, in a prospective cohort study by Parry *et al*, they found that 23–32% of people with knee OA reported significant pain variability.[5]

Pain in knee OA may be influenced by a range of other factors including levels of physical activity.[6 7] Like pain, physical activity assessment is also typically measured infrequently. Furthermore, it is often reliant on questionnaires, which are subjected to poor recall and poor precision. Understanding the relationship between pain and physical activity in knee OA has been limited by the ability to measure both variables simultaneously, accurately and in real time. The use of objectively measured levels of physical activity has been recommended,[8 9] yet only three known longitudinal studies have assessed the relationship between pain and steps per day in people with knee OA using such measures. One used an accelerometer to measure step count[10] and the other two used a research-grade activity monitor (StepWatch; ActivPAL).[11 12] However, in all three studies, pain levels were measured at different time points, in other words not simultaneously, to the physical activity.

Consumer technology including smartphones and wearables provides the opportunity to measure both patient-reported outcomes (such as pain) and passively collected sensor data (such as physical activity) contemporaneously. Smartphone apps are increasingly being used to track symptoms including in rheumatic diseases.[13] Although smartphones contain accelerometers allowing assessment of movement, they are not habitually carried by the user, and may thus misrepresent true patterns of activity.[14] Dedicated wearable devices such as fitness trackers, often wrist-worn, can continuously track movement, but rarely support concurrent data entry.

In this observational feasibility study, called 'Knee OsteoArthritis: Linking Activity and Pain' (KOALAP), we developed a cellular smartwatch app to record and assess pain in real time while also collecting raw and processed sensor data allowing contemporaneous capture of physical activity. The primary aim of the feasibility study was to test the feasibility, acceptability and ongoing engagement with smartwatch data collection for research.[15] In this analysis of data from the KOALAP study, we sought to examine patterns of pain and physical activity in people aged over 50 years living with OA. The specific objectives were:

1. To describe the day-to-day and within-day variability of pain intensity in people with symptomatic knee OA.
2. To assess how daily pain scores compare with pain assessed using the monthly Knee Injury and Osteoarthritis Outcome Score (KOOS) questionnaire pain domain scores.
3. To identify clusters of people with similar longitudinal pain scores.
4. To assess whether daily step count levels, assessed using the sensor data, are correlated with daily pain levels.

## METHODS

### Study design and sample

KOALAP was a longitudinal, observational feasibility smartwatch study in people with self-reported symptomatic knee OA. The full protocol for the KOALAP study has been published elsewhere[15] and there were no deviations from this protocol. In July 2017, the study was advertised in local newspapers and magazines and via social media channels. Participants had to be aged 50 years or above, have a self-report diagnosis of knee OA and be living in the Greater Manchester area or willing to travel to Manchester. Interested participants were asked to contact the study team, after which they were sent a patient information sheet and an invitation to one of four enrolment events. Participants were recruited to the study in September 2017 and data collection was completed in January 2018.

### Sample size

A minimum sample size of 20 participants was required based on expected attrition in order to describe patterns of longitudinal engagement, the primary aim of this feasibility study.[15 16]

### Data collection

At the enrolment event, participants self-reported their age, gender, height and weight. They were then provided with a consumer cellular smartwatch (Huawei Watch 2) with a bespoke app that collected (1) patient-reported outcomes via questionnaires and (2) continuous watch sensor data. The participants were instructed to wear the watch from waking until going to bed and to answer the watch questions when prompted within a specific time window, illustrated in table 1. All data were collected daily for 90 consecutive days.

For the two times per day 'level of knee pain' questions, participants were asked to record their level of knee pain on a 0–10 numeric rating scale.[17] To reduce responder burden, the KOALAP app used a subset of 26 items from the KOOS questionnaire in the app. This included the subscales on 'pain' and 'function, activities of daily living' (items P1–P9 and A1–A17, respectively, illustrated in the online supplemental table A1). These subscales were felt to contain the items most directly relevant to pain and function.[18] For this study, we decided to only include data from the 'pain' subscale (ie, questions P2–P9 only). Participants were asked to score their responses on the following five-point scale: none (0), mild (1), moderate

**Table 1** Summary of the frequency, trigger times and completion window times of the smartwatch questions

| Question | Frequency | Trigger times | Completion window times |
|---|---|---|---|
| Level of knee pain | Two times per day | 12:22 and 18:22 | 4 hours |
| 26 items from the Knee Injury and Osteoarthritis Outcome Score (KOOS) questionnaire | Monthly | Days 14, 44, 74 from start point | 1 week |

(2), severe (3) and extreme (4). We then took the sum of these item scores for each participant to give them a score for each month. We then scaled the score to a percentage, given the maximum possible score was 32, and divided by 10 to give us a score in the range between 1 and 10. We did not impute any missing data.

## Statistical methods

Descriptive statistics were used to describe participant characteristics including mean and standard deviation (SD) or median and interquartile range (IQR) for continuous data, and frequency (%) for categorical variables. Body mass index (BMI) was calculated from participants' height (m) divided by weight (kg) squared.

### Questionnaire data

We plotted two times per day pain scores over the 90-day period on individual graphs for each of the participants. On assessing data quality descriptively prior to the main analysis, we believed certain scores of 0 may have been submitted by accident. This was due to the user interface of the smartwatch app having a default value of zero and thus occasionally generating zero values as an error in data input. Given this error in the data input, all pain scores of 0 (of which there were 212 in total) were treated as missing values. Individual daily pain scores (numerical rating scale 1–10) were replotted, then visually reviewed for distribution patterns. The difference between early evening and lunchtime pain scores for each day of the 90-day period was calculated to assess within-day variability in level of knee pain. Monthly KOOS pain scores for each participant were plotted on top of the day-to-day pain scores.

The mean and variance of the lunchtime and early evening pain scores were calculated for each participant over the whole study period, giving an indication of the average level of pain and its variability within each participant's time series. These participant-level summary statistics were passed into a k-means algorithm.[19] From the plotted graphs of trajectories of daily pain scores for each of the participants, three broad patterns of pain were identified: sustained high levels of pain, sustained low levels of pain and fluctuating levels of pain. Based on this, clusters (k=3) were chosen for the algorithm; this grouped the participants into the three respective pain level groups. This was done separately for both the lunchtime and early evening pain; as the patterns of pain were similar, we decided to present only the lunchtime pain scores. In line with previous literature by Carlesso *et al* who have written extensively about the idea of OA causing two types of pain: intermittent (variable) and constant (stable),[20] we also explored a two-class model (based on variance only), which included constant (stable) pain and intermittent (variable) pain.

### Sensor data

Physical activity levels were assessed as total steps per day, measured by the app. The step counts were estimated from the accelerometry data using a proprietary algorithm. For our analysis, only estimated step counts over each calendar day were required. However, the data from the smartwatch were not recorded as daily step counts; but rather a monotonically increasing number sequence that resets at irregular intervals, corresponding to times when the smartwatch was recharged. To convert these data into daily step counts, we calculated the consecutive differences (ie, the increases from one data point to the next recorded data point, for that person) of the raw counts (ie, counts as recorded by the smartwatch) for each user. Where the raw differences were negative (ie, when the raw count has reset indicating the internal stepometer has reset) or the time lag between consecutive data points were more than 1 day, the difference was reset to zero. Resetting differences over more than 1 day to zero is important, because it is not known if the step count after such a gap is due to the accumulation of multiple days, or whether they were all from that single day. Therefore, it avoids spuriously high step counts for certain days. The sum of this modified sequence was then computed for each calendar date for each participant, giving the estimated daily step counts. We looked then at the mean and variance of step counts by pain cluster group. We also looked at the correlation between same day morning pain scores and step counts by participant using Pearson correlation.

All the statistical analyses were carried out using R V.3.6.0.[21] The R packages dplyr and tidyr[22] were used for data manipulation and the package ggplot2[23] was used to produce the graphs. The R code used for analysis is included as an online supplemental file.

### Missing data

We did not impute missing data as missing data may have not been missing at random. A patient may have stopped responding to the study either because they felt they do not have any symptoms worth reporting, or because they are in so much discomfort that using the app may have been a burden.

### Patient and public involvement

None.

## RESULTS

### Participant characteristics

A total of 26 participants were recruited. One participant left the study early and was excluded from the analysis. The mean age of the participants was 65 years, (SD=8 years). Of the 25 participants, just over half (n=13) were men. The median (IQR) BMI was 27 (25–35) kg/m$^2$.

### Daily knee pain

The two times per day questions were completed on around half of the participant days.[16] In total, there were 2756 pain reports (out of a potential of 4680 pain reports), giving an average completion rate of approximately 59%

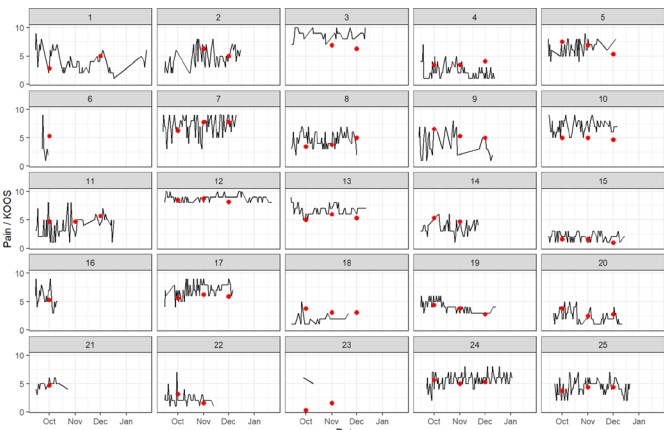

**Figure 1** Daily pain scores (assessed at 12:22) and monthly Knee Injury and Osteoarthritis Outcome Score (KOOS) questionnaire scores (represented by the red dots), by participant.

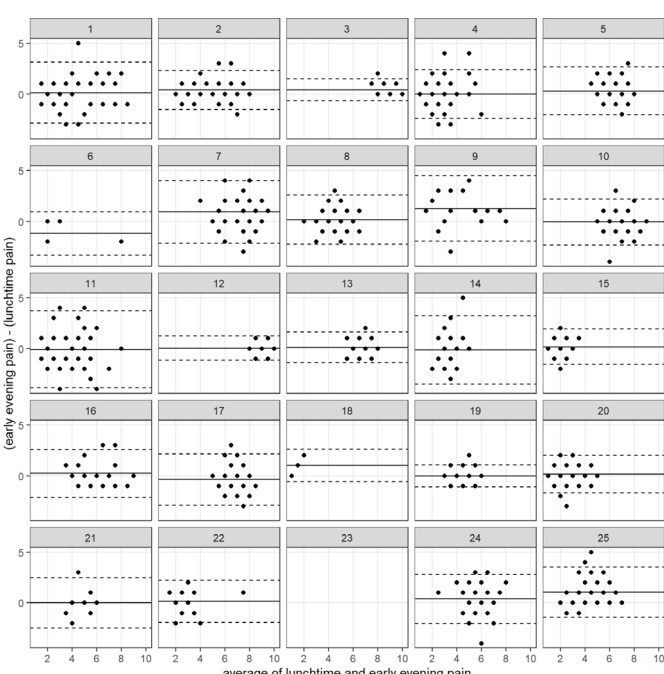

**Figure 2** Bland-Altman plot of self-reported daily scores for overall early evening versus overall lunchtime pain, by participant. The black line is the mean difference, the dashed lines are ±1.96 times SD (difference).

for these questions. Of these, 78 were 0 values, which were then excluded from the analysis given the concern about false entries (see methods section and online supplemental figure A1). Day-to-day patterns of knee pain, assessed at 12:22, for each of the 25 participants over the 3-month study period is presented in figure 1. The data show that there is a considerable amount of variation in the day-to-day levels of knee pain in the participants. Some participants (eg, participant IDs 3 and 12) had sustained high levels of pain, some participants (eg, participant ID 18) had sustained low levels of pain, while others (eg, participant IDs 1, 2, 9, 11) had more fluctuating pain levels.

For the majority of participants (n=20), there was no overall difference between their daily 12:22 and 18:22 pain scores (median difference of 0 (IQR 1)). In the remaining five participants, four reported worse pain in the early evening (participant IDs 9, 16, 24, 25) as illustrated graphically in figure 2, which shows a Bland-Altman plot of self-reported daily scores for overall early evening versus overall lunchtime pain, by participant. For most participants, the mean difference is near zero. There are some users for whom early evening pain scores are slightly higher (eg, participant ID 9).

As there was no difference in daily pain scores between the two time points, in the future analysis included in this manuscript, we focus only on the lunchtime pain scores.

## Monthly KOOS pain scores

The monthly KOOS pain scores (where available) for each participant were plotted alongside their day-to-day levels of pain, as red dots, in figure 1. It is not possible to directly compare the KOOS pain score and the daily level of pain; however, there were some interesting observations seen. For example, the KOOS pain scores were sometimes lower than that suggested by the daily pain scores (eg, in participant IDs 3 and 10) and sometimes the KOOS pain scores were higher than the daily pain scores (eg, in participant IDs 4 and 18). For most of the

participants the degree of variability in the KOOS pain scores within an individual was less marked than the day-to-day variability in pain though apart from perhaps one or two participants (eg, participant ID 3) tended to follow the broad pattern of daily pain over the 3 months.

## Cluster analysis

The mean and variance of the lunchtime (as well as the early evening) pain scores for each of the participants are shown in the online supplemental table A2. Using the k algorithm and including k-3 subjects were characterised into clusters (see figure 3). For each participant ID, we calculated group-wise summary statistics: namely the sample mean of the lunchtime pain and respectively of the early evening pain scores and the sample variances of the same. Hence, each patient is represented as a vector of four components: the mean and SD of the lunchtime pain and of early evening pain. These vectors are passed into the 'kmeans()' function from the base R stats package, with k=3 with 10 random restarts and the default Hartigan-Wong algorithm selected. The clusters described are the output from this function. On review of the lunchtime pain patterns (as shown in figure 1), it suggested three broad clusters of pain: high, low and fluctuating levels of pain. We repeated the clustering having removed participants who dropped out in the first half of the study as a sensitivity analysis and found the same result.

As mentioned in the methods section, we also explored a two-class model (based on variance only) which included constant (stable) pain and intermittent (variable) pain. Our findings of this are illustrated in the

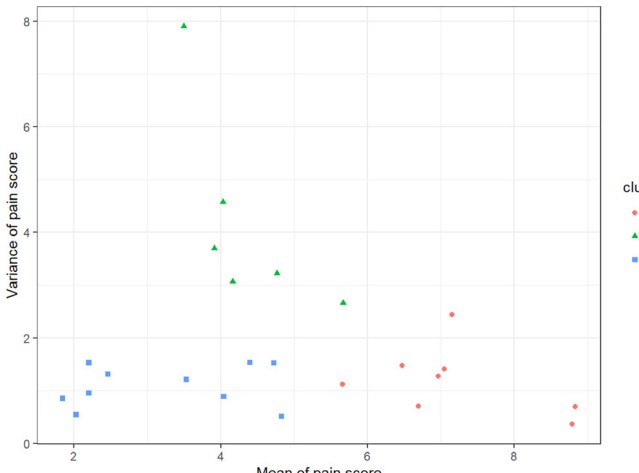

**Figure 3** Variance of lunchtime pain by mean pain score, with each point representing a pain cluster. The red circles (cluster 1) represent users with sustained high levels of pain, the blue squares (cluster 3) represent those with sustained low levels of pain and the green triangles (cluster 2) represent those with fluctuating pain levels.

online supplemental figure A2. We decided to focus on the three-cluster model in this paper and present the above methods in the main paper.

### Physical activity/step count data

The estimated daily step count levels for each of the 25 participants over the 3-month period is shown in figure 4. The figure highlights the variability in step counts with some participants having little variability (eg, participant IDs 11 and 12), others having high variability (eg, participant IDs 24 and 25) and one (participant ID 4) having two very distinct patterns of activity.

To explore whether the differences in step count levels between the participants were linked to their daily pain levels, we calculated the mean (and SD) step count per day for the three clusters, as shown in table 2. We found that all clusters had low levels of physical activity, especially the fluctuating pain levels cluster. Those with sustained high or sustained low pain levels had a similar step count level; however, those with fluctuating pain levels (ie, unpredictable levels of pain) had much lower step count levels.

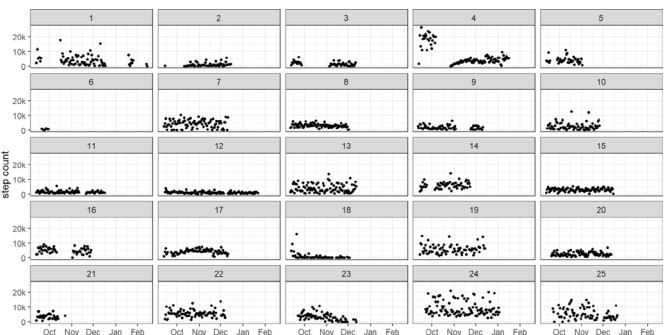

**Figure 4** Daily step count levels by participant, over the 3-month study period.

**Table 2** Mean and standard deviation (SD) of daily step count by pain cluster

| Pain cluster | Mean | SD |
|---|---|---|
| Sustained high pain levels | 3754 | 2524 |
| Sustained low pain levels | 4307 | 2992 |
| Fluctuating pain levels | 2064 | 1716 |

In an individual analysis, for the majority of participants (n=17) the correlation coefficients were positive indicating that increased step count was linked with increasing pain; however, the correlation coefficients were small, confidence intervals were wide and embraced unity, as illustrated in the online supplemental table A3. In some participants who had sustained high lunchtime pain scores, step count levels were low (eg, participant ID 3 and 12). However, other participants who had sustained low lunchtime pain scores, also had low step count levels (eg, participant ID 15). In those participants who had fluctuating pain scores (ranging from low to high), some had fluctuating step count levels (ranging from 0 to >20 000 steps) (eg, participant IDs 1, 24, 25), whereas others had sustained low step counts (eg, participant IDs 2, 9, 11) irrespective of high or low pain scores, as shown in figure 5.

### DISCUSSION

In our study, among people with knee OA we found that there was little within day but significant between day variability in pain. We found that when we categorised pain into three groups, patients separated into intermittent (variable) and constant, either sustained high or low

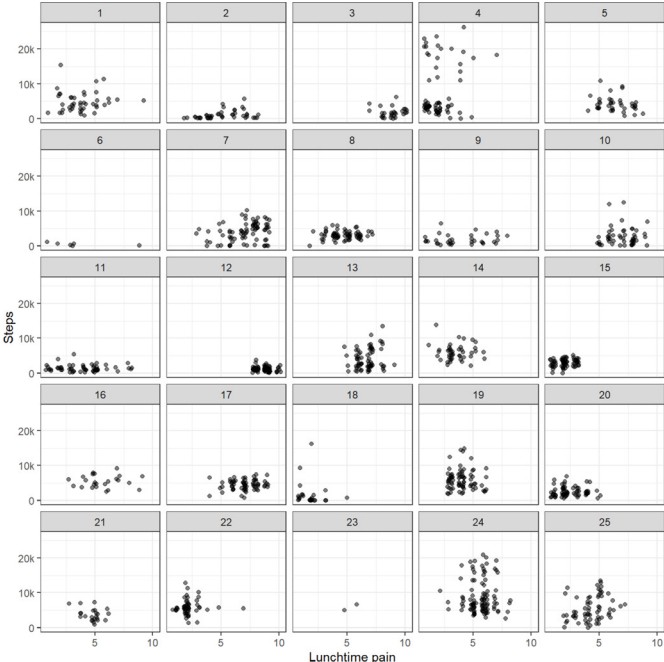

**Figure 5** Jittered scatter plot of daily step count level against overall lunchtime pain, by participant.

levels of pain, supporting what is already known.[7 24] We also found that there was no difference in pain scores between the lunchtime and early evening scores in most of our participants though it may be that the app timings (notifications at 12:22 and 18:22) were not truly representative of the maximal possible range of pain from morning to evening. Our study also found that participants with seemingly stable monthly KOOS pain scores had a lot of day-to-day variability in pain levels in between their monthly scores.

Our study also uniquely found that those with either sustained high or low levels of pain have a similar step count average but those with fluctuating pain have much lower levels of activity. This may indicate that fluctuating pain (ie, pain that is unpredictable/ always changing) may be more troublesome to people living with knee OA than those with, for example, sustained high levels of pain. Our finding that low levels of pain were not necessarily associated with higher step count levels highlights that there may be other factors related to low physical activity levels in people with knee OA (not just pain). For example, White *et al* found that people with depressive symptoms at baseline walked fewer steps per day.[11] It is also important to note the possible bidirectional nature of pain and physical activity, whereby low pain may enable greater activity, but also where increased activity may precipitate higher pain. Our analysis of this feasibility study did not examine this in detail, yet it provides evidence this might be possible in the future given the ability to collect time series data.

Studies to date remain ambiguous regarding the link between the pain in knee OA and physical activity, with some studies finding a positive association[6] and some studies no association.[10] These differences in findings may in part be due to the different methods used to measure physical activity and the timings when the level of pain was assessed. For example, in a study by Parry *et al*, paper diaries were used to collect information on daily pain in people with knee OA, to assess the impact of specific physical exposures on increases in daily pain levels.[6] However, as these paper diaries were completed retrospectively, they may have been subject to errors of inaccurate recall. With the use of this novel method of using consumer devices to measure symptoms and step count concurrently, it may help provide a more accurate assessment of these associations in the future.

There are, however, also several limitations to be considered when interpreting the findings from this study. This was a feasibility study and so the participant numbers were relatively small and biased with non-random missingness, so any conclusions drawn may not necessarily generalise to all future users of such an app or all people with knee OA. Subjects volunteered for participation, and it is possible that selection factors and in particular an interest in smartwatches may have influenced engagement and reporting, limiting the generalisation of the findings. Participants self-reported their knee OA; this diagnosis was not confirmed with their medical records,

we do not know if they are receiving any treatments, we did not have information on disease duration and thus participants may have had varying severities of their knee OA (early disease to end-stage). Moreover, due to the user interface of the smartwatch app having a default value of zero and thus occasionally generating zero-values as an error in data input, all zero values were treated as missing values to avoid 'false' zeros; we recognise that in doing this we may have also removed some 'true' zeros.

There are also several limitations related to the step count. The watch measured accelerometry data and a proprietary algorithm estimated the number of steps from these data. However in a study that compared commercially available activity tracking devices for step count accuracy, no significant differences were found among the devices.[25] The step count used was an estimate of the true step count levels, which were broken up by charging cycles- some activity may have been missed, leading to measurement error. Also, the total step count is a very rough measure of physical activity and misses patterns within the day.[25] Moreover, the qualitative details, such as whether the walking was slow/fast (which have stronger associations with pain), was missed. We do, however, have the raw accelerometer data and will conduct future analysis to help us to better understand the relationship between pain and more granular patterns of physical activity. Our analysis describes three clusters of longitudinal pain patterns and their physical activity. The descriptive analysis was exploratory rather than intending to formally compare these groups given the low sample size. This could be done in future studies.

It is well known that physical activity and exercise play a critical role in the conservative management of knee OA;[26] however, physical activity-based exposures may also trigger acute flares. The findings from this study support the notion that a one size fits all management of pain and physical activity in knee OA may not be effective and that a more personalised approach may be needed. In the future, the use of consumer devices may aid this personalised approach as it will help to identify the optimal level of physical activity for each patient before it triggers an exacerbation of pain. This opens the opportunity for personalised (digital) coaching, where patients might be guided to do a level of physical activity that is appropriate for them, encouraging activity levels up to, but not exceeding, a level that exacerbates pain. In summary, this feasibility study has shown that the use of the consumer cellular smartwatch app was successful in assessing and recording data on patient-reported outcomes and continuous sensor data simultaneously in real time. This data allowed us to highlight the substantial variability in knee pain due to OA and its links with physical activity. Larger studies are required to confirm our findings and allow exploration of the complex bidirectional relationships between activity and pain. This method of data capture should help to develop personalised physical activity recommendations for people with knee OA in the future,

based on what their optimal level of physical activity is in relation to their pain.

## Author affiliations

[1] Centre for Epidemiology Versus Arthritis, University of Manchester, Manchester, UK
[2] Nuffield Department of Orthopaedics, Rheumatology and Musculoskeletal Sciences, University of Oxford, Oxford, UK
[3] Department of Rheumatology, Oxford University Hospitals NHS Trust, Oxford, UK
[4] NIHR Manchester Biomedical Research Centre, Manchester University NHS Foundation Trust, Manchester Academic Health Science Centre, Manchester, UK
[5] Department of Rheumatology, Salford Royal NHS Foundation Trust, Salford, UK

**Acknowledgements** This project has been possible through collaboration with the Google Fit Research team at Google UK. The Google team has collaboratively built the Knee OsteoArthritis: Linking Activity and Pain app for self-reported data collection and the system for collecting and transmitting sensor data.

**Contributors** WD, TWO'N, JCS and MJP were involved in the conception, design and data collection for the study. DS and AV were involved in the data analysis and all authors (AV, DS, WD, TWO'N, JCS, MJP and ML) were involved in the interpretation of the results. AV drafted the article with critical revision from all authors (DS, WD, TWO'N, JCS, MJP and ML). All authors (AV, DS, WD, TWO'N, JCS, MJP and ML) approved the final version of the article to be published. WD acted as the guarantor.

**Funding** The study was funded by Versus Arthritis (grants 21225 and 21755) and supported by the NIHR Manchester Biomedical Research Centre (grant number N/A). AV is supported by a National Institute for Health and Research (NIHR) funded Academic Clinical Fellowship (grant number N/A).

**Competing interests** WD has received consultancy fees from Google and Abbvie.

**Patient and public involvement** Patients and/or the public were not involved in the design, or conduct, or reporting, or dissemination plans of this research.

**Patient consent for publication** Not applicable.

**Ethics approval** This study involves human participants. The study underwent full review by the University of Manchester Research Ethics Committee (#0165). Participants gave informed consent to participate in the study before taking part.

**Provenance and peer review** Not commissioned; externally peer reviewed.

**Data availability statement** No data are available.

**ORCID iDs**
Arani Vivekanantham http://orcid.org/0000-0003-4605-6598
Matthew J Parkes http://orcid.org/0000-0002-1574-9933

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
