## [Reviewer comments · BMJ Open]

ARTICLE DETAILS

TITLE (PROVISIONAL)	Day to day variability of knee pain and the relationship with physical activity in people with knee osteoarthritis: an observational, feasibility study using consumer smartwatches
AUTHORS	Vivekanantham, Arani; Selby, David; Lunt, Mark; Sergeant, Jamie; Parkes, Matthew; O'Neill, Terence; Dixon, Will

VERSION 1 – REVIEW

REVIEWER	Jason Wallis La Trobe University, Allied Health
REVIEW RETURNED	10-May-2022

GENERAL COMMENTS	Peer Review -This study assessed the feasibility of using an App and Smartwatch to capture pain and physical activity in real time. This is an important study assessing a novel and accessible way to capture this data, that may ultimately help to better understand both pain and physical activity in people with this condition. It could help to develop new strategies to increase/optimize physical activity levels which is such an issue for people with knee OA. I have made some comments below that are designed to improve the submission. Title -As this is a feasibility study, the design 'feasibility' is missing from the title. Abstract (Objective) -The objective was "to conduct a feasibility study...". However, it would read better if the aim was "To assess the feasibility of...." Or to explore the feasibility of..." Abstract (Design)/Methods (Design) -In both sections, need to include the study design, and clarify how feasibility was assessed. Including justification for the feasibility measures would be helpful as well. -In the methods can you include reporting guidelines for this feasibility study? Results Looking at Table 2, the mean physical activity levels suggest all clusters had low levels of physical activity, especially the fluctuating pain levels cluster - that appears very low. I think this needs to be mentioned. Discussion -Para 1 - Consider replacing the word "successful" with "feasible" and why. i.e. based on methods on how feasibility was assessed which is currently unclear in the methods. -Para 3 – Not sure what para 3 adds, or how this links to this
---

	feasibility study's findings that are included in para 2 and 4. Perhaps an explanation of the key finding in para 2 is needed to explain the variable nature of pain day-to-day, but not within-day variability. -Para 4 – How does this key finding (“those with either sustained high or low levels of pain have a similar step count average but those with fluctuating pain have much lower levels of activity”) compare/contrast with the 3 previous studies cited in the abstract – ref 10, 11, 12? Is this a unique finding? Conclusion I don't think this feasibility study's findings give rise to making clinical recommendations in the conclusion. i.e I would consider omitting “Therefore, a personalized approach to physical activity in knee OA is recommended, adjusted to the patient's phenotype” from the conclusion. Instead these clinical and research implications could be included earlier in the discussion. I think the findings support the notion that a one size fits all approach to management of pain and increasing physical activity may not be effective. Perhaps could give an examples of how to personalise the approach based on the findings as an area of further research, particularly for the cluster with the lowest level of physical activity as this group has the most to gain.
--	---

REVIEWER	Ebru Kaya Mutlu Istanbul Universitesi, Physiotherapy and Rehabilitation
REVIEW RETURNED	10-May-2022

GENERAL COMMENTS	All my comments are in the textboxes at each point of the manuscript.  - The rationale for the study is not convincing, we recommend that the introduction be strengthened. The introduction was too long (especially first and second paragraph) and was not clearly focused. What is the current evidence for the use Day to day variability of knee pain for knee OA? I cannot view the current study in ClinicalTrials.gov. Please ensure to register your study in Clinical Trials and include a ClinicalTrials.gov Identifier number in the study design. - There is insufficient information about the patients. For example; WHEN DID THEY RECEIVE THE DIAGNOSIS OF OA? or Are the patients receiving treatment? - In addition to the scientific output of the article, I think that the 29-person data document is over-interpreted. Therefore, type-1 and type 2 errors should be reviewed by the statistician.
--

REVIEWER	Sezgin Ciftci Temple University College of Public Health, Epidemiology and Biostatistics
REVIEW RETURNED	31-Aug-2022

GENERAL COMMENTS	Title: Day to day variability of knee pain and the relationship with physical activity in people with knee osteoarthritis: A study using consumer smartwatches. Thank you for providing the opportunity to be a peer reviewer for this interesting feasibility study about the variability of knee pain and its relationship with daily walking activities in people with knee osteoarthritis (OA) by using smartwatches. The focus of this feasibility study is a sample of 25 participants aged more than 50 years with symptomatic OA. The knee pain scores and step counts are recorded by using a smartwatch and an app. Then the relationship between daily step counts and categorized knee pain
---

	levels are explored by using correlations measures and clustering the participants according to their pain variability. The results with some limitations are discussed, and it is concluded that smartwatches can be a source to assess the pain level and the physical activity in knee OA. As a peer reviewer, I only focused on the statistical methods and the analysis conducted. Here are my comments for this study:  • More evidence related to the relationship between step counts and pain score needs to be given. Table A3 includes only the individual coefficients without any test of significance. More detailed analysis is needed such as correlation coefficients for each participant, for each cluster and in total with necessary statistically significance tests. Moreover, results of Figure 5 need to be discussed clearly for more evidence about the relationship. • In Figure 1, it seems that the dataset contains lots of dropouts or missing values. For participants with ID 6, 14, 16, 18, 21, 22 and 23, the dropouts had happened before the half of the given time period. Moreover, as it is stated between the lines 215 – 217 at the Results section, the missing rate for the pain reports is 49%. Accordingly, the missingness mechanisms needs to be discussed in the context of the study and the effect of missingness needs to be discussed by conducting a proper sensitivity analysis. I personally suggest analyzing the data without those participants mentioned above and check if there is any difference with the current results. In addition to that, a justification of no imputation to the missing values should also be indicated. • In the methodology section and “Cluster analysis” subsection, the method used for clustering (k- means algorithm) needs to be explained within the context of the study (i.e., how it is implemented in this study). • It would be nice to share the R codes used for the analysis as a supplementary material in order to understand the methodology and the analyses better. Recommendation: Revision and re-submit
--	---

VERSION 1 – AUTHOR RESPONSE

Reviewer: 1

Dr. Jason Wallis, La Trobe University, Cabrini Health Comments to the Author:

Peer Review

-This study assessed the feasibility of using an App and Smartwatch to capture pain and physical activity in real time. This is an important study assessing a novel and accessible way to capture this data, that may ultimately help to better understand both pain and physical activity in people with this condition. It could help to develop new strategies to increase/optimize physical activity levels which is such an issue for people with knee OA. I have made some comments below that are designed to improve the submission.

Thank you for your comments to improve our submission.

Title

-As this is a feasibility study, the design ‘feasibility’ is missing from the title.

Thank you for your feedback. I have now updated the article title.

Abstract (Objective)

-The objective was “to conduct a feasibility study...”. However, it would read better if the aim was “To assess the feasibility of....” Or to explore the feasibility of...”

Thank you for your feedback. I have now amended this.

Abstract (Design)/Methods (Design)

-In both sections, need to include the study design, and clarify how feasibility was assessed. Including justification for the feasibility measures would be helpful as well.

-In the methods can you include reporting guidelines for this feasibility study?

Thank you for your comment. I have now included the study design (observational, feasibility study) in the abstract and methods section. The feasibility study was focussed on whether participants would remain engaged with a smartwatch study that requested this frequency of data over a three-month period. The results of this analysis are reported elsewhere (ref Beukenhorst et al, reference 15). In this publication, we report preliminary data on the clinical findings derived from this approach of collecting daily data in OA. We have now removed the first paragraph of the discussion section as assessing feasibility was not the primary aim of this analysis. We have also made the differentiation clearer at the end of the introduction as mentioned in response to the Editor's comments.

Results

Looking at Table 2, the mean physical activity levels suggests all clusters had low levels of physical activity, especially the fluctuating pain levels cluster - that appears very low. I think this needs to be mentioned.

Thank you for your feedback. I have now mentioned this in the paragraph above Table 2 on page 17.

Discussion

-Para 1 - Consider replacing the word "successful" with "feasible" and why. i.e. based on methods on how feasibility was assessed which is currently unclear in the methods.

Thank you for your feedback. As mentioned in my response to the comment about feasibility above, we are not formally assessing feasibility within this analysis. The feasibility study was focussed on whether participants would remain engaged with a smartwatch study that requested this frequency of data over a three-month period. The results of this analysis are reported elsewhere (ref Beukenhorst et al, reference 15). In this publication, we report preliminary data on the clinical findings derived from this approach of collecting daily data in OA. We have now removed the first paragraph of the discussion section as assessing feasibility was not the primary aim of this analysis and hope this helps clarify this.

-Para 3 – Not sure what para 3 adds, or how this links to this feasibility study's findings that are included in para 2 and 4. Perhaps an explanation of the key finding in para 2 is needed to explain the variable nature of pain day-to-day, but not within-day variability.

Thank you for your feedback. I have now deleted paragraph 3. In paragraph 2 I have explained the variable nature of day-to-day pain, noting also that there is limited within-day variability.

-Para 4 – How does this key finding ("those with either sustained high or low levels of pain have a similar step count average but those with fluctuating pain have much lower levels of activity") compare/contrast with the 3 previous studies cited in the abstract – ref 10, 11, 12? Is this a unique finding?

Thank you for your comment. Yes, this is a unique finding from this study. Both reference 12 (Parkes et al) and 10 (Brisson et al) found that pain and activity were not associated. Reference 11 (White et al) found that people with depressive symptoms at baseline walked fewer steps/day, and there was a trend for people with radiographic OA worsening to walk fewer steps/day. I have now added this to paragraph 4.

Conclusion

I don't think this feasibility study's findings give rise to making clinical recommendations in the conclusion. i.e I would consider omitting "Therefore, a personalized approached to physical activity in knee OA is recommended, adjusted to the patient's phenotype" form the conclusion.

Thank you for your feedback. I have now omitted this.

Instead these clinical and research implications could included earlier in the discussion. I think the findings support the notion that a one size fits all approach to management of pain and increasing physical activity may not be effective. Perhaps could give an examples of how to personalise the approach based on the findings as an area of further research, particularly for the cluster with the lowest level of physical activity as this group has the most to gain.

Thank you for your comment. I have now amended the discussion (second to last paragraph) to include this. We have added the following text to provide the examples that you suggest:

“This opens the opportunity for personalised (digital) coaching, where patients might be guided to do a level of physical activity that is appropriate for them, encouraging activity levels up to, but not exceeding, a level that exacerbates pain.”

Reviewer: 2

Dr. Ebru Kaya Mutlu, Istanbul Universitesi Comments to the Author:

All my comments are in the textboxes at each point of the manuscript.

- The rationale for the study is not convincing, we recommend that the introduction by strengthened. The introduction was too long (especially first and second paragraph) and was not clearly focused. What is the current evidence for the use Day to day variability of knee pain for knee OA?

Thank you for your feedback. We have re-read the introduction and believe the background is relevant and necessary to justify our study and analysis. The other reviewers did not have any concerns about the introduction length. We have therefore not shortened the introduction but would be happy to revisit if the editor felt this necessary. We have nonetheless tried to describe more clearly the evidence for the day-to-day variability in knee pain by referencing examples from published studies (e.g., Parry et al, reference 4).

I cannot view the current study in ClinicalTrials.gov. Please ensure to register your study in Clinical Trials and include a ClinicalTrials.gov Identifier number in the study design.

Thank you for your comment. We published the study protocol (reference 15), but not on ClinicalTrials.gov.

- There is insufficient information about the patients. For example; WHEN DID THEY RECEIVE THE DIAGNOSIS OF OA? or Are the patient receiving treatment?

Thank you for your comment. The patients had a self-report diagnosis of knee OA. We do not have any details on when they received the diagnosis of OA and whether they are receiving treatment. We have made this more explicit by stating this in the first paragraph of the methods section and also in the limitations in the discussions section.

- In addition to the scientific output of the article, I think that the 29-person data document is over-interpreted. Therefore, type-1 and type 2 errors should be reviewed by the statistician.

Thank you for your comment. We have reviewed this with our statistician. They explained that as Type 1 and Type 2 errors are a result of null hypothesis significance testing and this project is mostly descriptive and largely does not depend on significance tests. However, we agree it is worth emphasising that as the sample size is small and biased with non-random missingness, any conclusions drawn may not necessarily generalize to all future users of such an app, let alone all people with knee osteoarthritis. We have made this more explicit by adding the above to the limitations in the discussion section.

“This was a feasibility study and so the participant numbers were relatively small and biased with non-random missingness, so any conclusions drawn may not necessarily generalize to all future users of such an app or all people with knee osteoarthritis.”

Reviewer: 3 [PLEASE SEE ATTACHED FILE FOR COMMENTS FROM REVIEWER 3] Dr. Sezgin Ciftci, Temple University College of Public Health Comments to the Author:
My review and comments are attached.

More evidence related to the relationship between step counts and pain score needs to be given. Table A3 includes only the individual coefficients without any test of significance. More detailed analysis is needed such as correlation coefficients for each participant, for each cluster and in total with necessary statistically significance tests. Moreover, results of Figure 5 need to be discussed clearly for more evidence about the relationship.

Thank you for your feedback. Table A3 has now been updated to include 95% confidence intervals. I have added more detail about the results of Figure 5 to the last paragraph of the results section, pls added the following text in the discussion:

“Our analysis describes three clusters of longitudinal pain patterns and their physical activity. The descriptive analysis was exploratory rather than intending to formally compare these groups given the low sample size. This could be done in future studies.”

In Figure 1, it seems that the dataset contains lots of dropouts or missing values. For participants with ID 6, 14, 16, 18, 21, 22 and 23, the dropouts had happened before the half of the given time period. Moreover, as it is stated between the lines 215 – 217 at the Results section, the missing rate for the pain reports is 49%. Accordingly, the missingness mechanisms needs to be discussed in the context of the study and the effect of missingness needs to be discussed by conducting a proper sensitivity analysis. I personally suggest analyzing the data without those participants mentioned above and check if there is any difference with the current results. In addition to that, a justification of no imputation to the missing values should also be indicated.

Thank you for your feedback. The missing data may have not been missing at random. Patients may have stopped responding to the study either because they felt they do not have any symptoms worth reporting, or because they are in so much discomfort that using the app may have been a burden. We have made this more explicit in lines 226 – 230.

“We did not impute missing data as missing data may have not been missing at random. Patient may have stopped responding to the study either because they felt they do not have any symptoms worth reporting, or because they are in so much discomfort that using the app may have been a burden.”

All smartphone or smartwatch health studies collecting daily tracked symptom data have missing data. Our engagement patterns were good compared to much of the literature. The reviewer’s suggestion of dropping participants with missing data would mean we would need to drop all participants.

For the benefit of the reviewer, we analysed the data without the specific participants mentioned above and did not see any obvious difference with the current results (please see graph below). We have added a line to this effect in the results section as follows:

“We repeated the clustering having removed participants who dropped out in the first half of the study as a sensitivity analysis and found the same result.”

In the methodology section and “Cluster analysis” subsection, the method used for clustering (k-means algorithm) needs to be explained within the context of the study (i.e., how it is implemented in this study).

Thank you for your feedback. We have now explained the method used for clustering in the “cluster analysis” sub-section of the methods section (also included below).

For each participant ID we calculated group-wise summary statistics: namely the sample mean of the morning pain and respectively of the afternoon pain scores and the sample variances of the same. Hence each patient is represented as a vector of four components: the mean and standard deviation of morning pain and of afternoon pain. These vectors are passed into the “kmeans()” function from the base R stats package, with k = 3 with 10 random restarts and the default Hartigan–Wong algorithm selected. The clusters described are the output from this function.

It would be nice to share the R codes used for the analysis as a supplementary material in order to understand the methodology and the analyses better.

Thank you for your feedback and helpful suggestion. We have shared the R codes used for the analysis as a supplementary material.

VERSION 2 – REVIEW

REVIEWER	Jason Wallis La Trobe University, Allied Health
REVIEW RETURNED	18-Oct-2022
GENERAL COMMENTS	I am happy the authors have addressed all the concerns and questions I raised.

REVIEWER	Sezgin Ciftci Temple University College of Public Health, Epidemiology and Biostatistics
REVIEW RETURNED	02-Nov-2022

GENERAL COMMENTS	Title: Day to day variability of knee pain and the relationship with physical activity in people with knee osteoarthritis: An observational, feasibility study using consumer smartwatches. Thank you for providing the opportunity to be a peer reviewer for the revised version of this study. The authors have added more details on the correlations between step counts and pain scores and discussed them briefly. They have also discussed more missing data mechanisms and added extra analysis about the potential missing data problem to show that there is no observed change after taking out the subjects who left the study before half of the time period. In the methodology section, the authors have explained how the cluster analysis was applied within the context of the study. Lastly, they provided the codes for the analysis which helps a lot to understand the analysis and the results better. All my reviews and comments were considered and evaluated point by point by the authors. I don't have any other specific comments on this revised version. Recommendation: Accept
--